# The Potential of Bemegride as an Activation Agent in Electroencephalography in Dogs

**DOI:** 10.3390/ani12223210

**Published:** 2022-11-19

**Authors:** Junya Hirashima, Miyoko Saito, Minoru Yokomori

**Affiliations:** Laboratory of Small Animal Surgery (Neurology), School of Veterinary Medicine, Azabu University, Sagamihara 252-5201, Japan

**Keywords:** activation methods, bemegride, dog, electroencephalography, epilepsy, epileptiform discharge, irritative zone, sevoflurane

## Abstract

**Simple Summary:**

Electroencephalography is being increasingly recognized as invaluable in the diagnosis and management of epilepsy in veterinary medicine. However, the occurrence of seizures cannot be predicted, and abnormal findings are not always readily available at the time of examination. Activation methods are techniques that provoke interictal electroencephalogram abnormalities, such as epileptiform discharges (EDs), and are routinely used in clinical settings in human medicine. Only a few activation methods have been investigated in veterinary medicine, and their effectiveness remains controversial. Therefore, the present study investigated the potential of bemegride as a pharmacological activator in dogs. The obtained results demonstrated that bemegride activated interictal EDs under anesthesia in all dogs with epilepsy, seemingly at spontaneous irritative zones. The dose required for ED activation was significantly lower in dogs with epilepsy (median; 7.3 mg/kg) than in those without (median; 19.7 mg/kg) (*p* = 0.0294). Furthermore, there were no serious adverse effects, such as the induction of clinical seizures, associated with the administration of bemegride. Therefore, bemegride has potential as a safe and effective activation agent in dogs with epilepsy. The present results provide more options for the diagnosis and therapeutic planning of epilepsy, including presurgical evaluations, in dogs.

**Abstract:**

The present study investigated the potential of bemegride as a pharmacological activation agent that elicits epileptiform discharges (EDs) in interictal electroencephalogram (EEG) recordings in dogs. Four laboratory dogs with idiopathic epilepsy and four without epilepsy were included. The dogs were anesthetized using sevoflurane during EEG recordings. Bemegride was administered intravenously and repeatedly until EDs were enhanced or induced, or until the maximum dose (20 mg/kg) had been administered. Bemegride activated EDs in all dogs with epilepsy. These EDs predominantly occurred in each dog’s spontaneous irritative zones, which were identified without the administration of bemegride. EDs occurred after the administration of bemegride in 50% of dogs without epilepsy. The dose required for activation was significantly lower in dogs with epilepsy (median; 7.3 mg/kg) than in those without (median; 19.7 mg/kg) (*p* = 0.0294). The only suspected adverse effect associated with the administration of bemegride was vomiting in two dogs after awakening from anesthesia. There were no other adverse effects, including seizures. The present results demonstrated the potential of bemegride as a safe and effective pharmacological activation agent of EDs in anesthetized dogs with epilepsy and provided more options for the diagnosis and therapeutic planning of epilepsy, including presurgical evaluations, in dogs.

## 1. Introduction

Epilepsy is a functional chronic brain disease that causes epileptic seizures. It is one of the most common neurological disorders in dogs [1,2]. An epileptic seizure is caused by abnormal electrical activity in the brain, and electroencephalography (EEG) is the only modality that shows epileptogenicity and identifies the location of epileptogenic foci in the brain. The importance of EEG in the diagnosis and management of epilepsy has been emphasized in veterinary medicine [3,4] (pp. 153–181). In the International Veterinary Epilepsy Task Force (IVETF) consensus criteria, the identification of ictal or interictal EEG abnormalities is essential in Tier III, the highest confidence level for a diagnosis [3].

An epileptic seizure is a clinical manifestation of temporal electrical abnormalities inside the brain; therefore, the simultaneous evaluation of EEG and seizure-like activity in patients is the most effective approach for reaching a diagnosis. Long-term video-EEG monitoring, which comprises the simultaneous recording of EEG and patients’ movements, has played an important role in diagnosing, classifying, and quantifying seizures in human medicine [5]. In veterinary medicine, it is still challenging to perform EEG recordings in awake dogs for various reasons, including a lack of cooperation and muscle artifacts. Therefore, interictal EEG recordings in sedated or anesthetized dogs are commonly conducted in veterinary medicine [6,7,8,9,10,11,12,13].

The identification of epileptiform discharges (EDs) in interictal EEG recordings facilitates the diagnosis of epilepsy in dogs and humans [10,14]. Recent studies detected EDs in the interictal period and showed that they were more frequent in dogs with frequent clinical seizures or a severe episode (i.e., an episode of status epilepticus or cluster seizures) than in those without epilepsy, with less frequent clinical seizures, or without severe episodes [10]. Therefore, EEG may contribute to a diagnosis in dogs with frequent or severe epileptic seizure-like episodes. However, some dogs do not show EDs in interictal EEG recordings even though they have epilepsy, and the detection rate in dogs with epilepsy currently ranges between 20 and 86% [8,9,10,11,12,13].

Activation methods have been utilized in human medicine to improve the diagnostic yield for epilepsy. Activation methods enhance preexisting EEG abnormalities, such as EDs. Two types of activation methods have been reported in veterinary medicine: photic stimulation and hyperventilation [7,8,10,15,16]; however, they provide inconsistent findings. Neither hyperventilation nor photic stimulation provoked EDs in all epileptic dogs [8], while photic stimulation provoked EDs in 7 of the 13 epileptic dogs [15]. Another study that used photic stimulation did not describe whether EDs were elicited by the stimulation [16]. Therefore, to improve the diagnostic yield for epilepsy, the activation methods of EEG in veterinary medicine warrant further investigation.

Pharmacological activation is another activation technique that uses drugs to provoke interictal EEG abnormalities. Bemegride is one of the drugs used in human medicine [17,18,19,20,21] and has been shown to elicit EEG abnormalities, including EDs, in 67.5–82.7% of patients [19,20,21]. Experimental studies previously reported that bemegride provoked EDs in cats [22,23,24]. An anecdotal study administered bemegride to two dogs with epilepsy during interictal EEG recordings and showed that it provoked interictal EDs [25]. However, it currently remains unclear whether bemegride has potential as a pharmacological activation agent in dogs.

Therefore, the present study investigated the feasibility of bemegride as a pharmacological activation agent of interictal EEG abnormalities (particularly EDs) in dogs with epilepsy.

## 2. Materials and Methods

### 2.1. Animals

Eight laboratory dogs were included in the present study. Among them, four dogs (2 beagles, 1 miniature dachshund, and 1 Pekinese; three females and one male) had naturally occurring idiopathic epilepsy (IE), which was diagnosed based on Tier I IVETF consensus criteria in two dogs and Tier III criteria in another dog [3]. The diagnosis in the remaining dog was reached using Tier I criteria, except for urinalysis and serum NH_3_ levels. Two of the dogs had their diagnosis confirmed by necropsy after their deaths from diseases unrelated to epilepsy. The seizure type of these four dogs was focal motor epileptic seizures that evolved into generalized tonic-clonic seizures. These four dogs were defined as the epilepsy group in the present study. Two out of the four dogs in the epilepsy group had been routinely administered anti-seizure drugs. The mean age and weight of these dogs were 48.5 (range, 25–109) months and 7.65 (range, 3–12) kg, respectively. The other four dogs were healthy laboratory beagles (one female and three males) and were defined as the control group. The mean age and weight of the control group were 43 (range, 25–60) months and 10.15 (range, 9.3–10.45) kg, respectively. Information on the dogs is shown in Table 1. Seizure type, clinical seizure signs (semiology), and the laterality of seizure signs in each dog are shown in Table 2.

### 2.2. Anesthetic Procedure

A complete blood cell count (CBC), serum biochemistry tests, and chest radiography were performed within one week before EEG recordings in all dogs. General anesthesia was induced by administering 5.0% sevoflurane (Sevofulo^®^, DS Pharma Animal Health, Osaka, Japan) in oxygen via a mask until endotracheal intubation was possible. After endotracheal intubation, anesthesia was maintained with sevoflurane delivered in 100% oxygen. A semi-closed circle system was used. Dogs were positioned in sternal recumbency. Ventilation was maintained by spontaneous respiration. During anesthesia, the end-tidal CO_2_ concentration (EtCO_2_), peripheral hemoglobin saturation (SpO_2_), respiratory rate, electrocardiogram (ECG), and esophageal temperature were continuously recorded. ETCO_2_ and SpO_2_ were maintained at 30–40 mmHg and 95–100%, respectively. Temperature was maintained at 37.0–38.5 °C using hot water bottles. Lactated Ringer’s solution was administered intravenously at a rate of 10 mL/kg/h during EEG recordings.

### 2.3. EEG

A digital electroencephalograph (Neurofax EEG-9100, NIHON KOHDEN, Shinjuku-ku, Japan) was used to record EEG (time constant, 0.3 s; high-cut filter, 60 Hz). A subdermal needle electrode (stainless steel EEG needle, length of 22.5 mm, diameter of 0.22 mm, 1.5-meter cable; NE-224s, NIHON KOHDEN, Shinjuku, Japan) was used as the recording electrode. The placement of electrodes was the same as that reported by Utsugi et al. [10]. An electrode was placed symmetrically on both sides over the frontal (LF, RF), parietal (LP, RP), occipital (LO, RO), and temporal (LT, RT) lobes. Reference electrodes were placed on the left and right ears. A ground electrode was placed on the dorsal side of the neck above the atlas. Electrodes for recording eye movement were placed near the left and right eyelids (Figure 1). ECG was recorded using electrodes placed on both axillae. Bilateral ear reference electrode derivation and bipolar derivation were performed.

### 2.4. Procedure

Figure 2 shows the timetable for the administration of bemegride and EEG recordings. After endotracheal intubation, electrodes were placed subcutaneously on the head and EEG recordings were started. The concentration of sevoflurane decreased by 0.5–1.0% from 5.0% until burst suppression (BS) disappeared on the EEG trace, and was then maintained at this concentration throughout the remainder of EEG recordings. After BS disappeared, EEG was recorded for 20 min, and this period was defined as the baseline period. After the baseline period, 20 mg bemegride (Medibal^®^ injection 50 mg, Mitsubishi Tanabe Pharma Corporation, Osaka, Japan) was injected intravenously over 1 min in each dog regardless of its body weight (20 mg/head). An additional dose (20 mg/head) was administered every 2–3 min until EDs appeared on the EEG trace (i.e., until EDs were “induced”) [19]. If EDs were already present in the baseline period, the administration of bemegride was repeated every 2–3 min until an increase in the frequency of EDs was observed by visual impressions (i.e., until EDs were “enhanced”). The administration of bemegride was stopped when its dosage reached the maximum of 20 mg/kg [26] (p. 348). EEG recordings were continued for 20 min after the administration of the last dose of bemegride, and this period was defined as the activation period. When the activation period ended, diazepam (0.5 mg/kg; Horizon^®^ Injection 10 mg, Maruishi Pharmaceutical, Osaka, Japan) and phenobarbital (3 mg/kg; Phenobal^®^ Injection 100 mg, Fujinaga Pharma Corporation, Chuou-ku, Japan), which are bemegride antagonists [27,28], were administered intravenously. However, if an ictal EEG pattern appeared during the activation period, diazepam (0.5 mg/kg) and phenobarbital (3 mg/kg) were administered immediately. After confirming that EEG abnormalities were not observed for at least 5 min, dogs were awakened from anesthesia and constantly monitored for 24 h.

To evaluate the adverse effects of bemegride, CBC and serum biochemistry tests, in addition to physical and neurological examinations, were performed on the day after the experiment.

### 2.5. Analyses

EEG obtained in the baseline and activation periods were visually analyzed, with a focus on the frequency, type, and distribution of EDs in accordance with the previously reported method [10,24]. The type of ED was defined based on a previous study [10]. A spike or sharp wave with an amplitude ≥50 μV was counted [10]. In the present study, a series of polyspikes or rhythmic spikes were counted as one ED.

The irritative zone is the area of the cortex that generates interictal EDs [29] (pp. 137–159), [30]. The region in which EDs with the highest amplitude, rapid onset, and shortest duration by the referential derivation and/or with phase reversal by the bipolar derivation appeared most frequently was identified as the irritative zone. The number of EDs that appeared in the irritative zone was counted as the frequency of EDs (per minute). If EDs appeared to be completely synchronized and had the same amplitude at all electrodes, i.e., generalized EDs, the irritative zone was defined as generalized and the frequency of EDs was counted in one region. EEG reviewers were not blinded to the epilepsy/control status of the dog.

The irritative zone, the type of ED, and the frequency of EDs were compared between the baseline and activation periods. The total dose of bemegride was also compared between the two groups. The adverse effects of bemegride were assessed in each dog.

### 2.6. Statistical Analysis

Statistical analyses were performed using JMP version 7.0.1 (SAS Institute Japan Ltd., Minato-ku, Japan). The Wilcoxon signed-rank test was used to assess the significance of differences in the frequency of EDs between the baseline and activation periods in each group. The Mann–Whitney U test was performed to evaluate the significance of differences in the total bemegride dose between the epilepsy and control groups. A *p*-value < 0.05 was considered to be significant.

## 3. Results

Table 3 shows the regions identified as irritative zones. None of the dogs had generalized EDs.

In the epilepsy group, EDs appeared in three out of four dogs in the baseline period and these dogs also had EDs in the activation period. The region of the irritative zone was the same between the two periods. The typical EEG of a dog in the epilepsy group is shown in Figure 3. The remaining dog (dog 3) in the epilepsy group had no detectable EDs during the baseline period, but developed EDs after the administration of bemegride. In this dog, EDs were detected at a later date when EEG was repeated without bemegride. The region of the irritative zone identified with and without bemegride was consistent.

In the control group, EDs did not occur during the baseline period in all four dogs, but appeared in two dogs after the administration of bemegride.

Table 4 shows changes in ED wave types. Activated EDs appeared to have more types than baseline EDs.

The frequency of EDs increased in the activation period (Table 5). The median frequencies of EDs in the baseline and activation periods in the epilepsy group were 0.15 (range: 0–4.4, IQR: 0.075–1.25)/minute and 1.3 (range: 0.4–18, IQR: 0.85–5.7)/minute, respectively. These frequencies did not significantly differ (*p* = 0.125). None of the dogs in the control group had EDs in the baseline period, and the median frequency of EDs in the activation period was 0.1 (range: 0–0.4, IQR: 0–0.35)/minute. No significant differences were observed in the frequency of EDs between the two periods (*p* = 0.500) (Table 5).

The median total doses of bemegride required to enhance or induce EDs were 7.3 (range: 5.3–8, IQR: 6.35–7.925) mg/kg in the epilepsy group and 19.7 (range: 17.2–20, IQR: 18.85–20) mg/kg in the control group. These doses were significantly different without overlap (*p* = 0.0294) (Figure 4).

During the procedure, the frequency of EDs in one dog (dog 2) in the epilepsy group markedly increased based on visual impressions (from 4.4/min to 18/min) after the first administration of bemegride. Therefore, 0.5 mg/kg diazepam and 3 mg/kg phenobarbital were injected 6 min after the administration of bemegride, and EDs disappeared immediately. In this dog, the 6 min EEG after the administration of bemegride was assessed as the activation period. No dogs showed ictal EEG patterns during the procedure.

Changes in respiration patterns were noted after the administration of bemegride in three dogs: one in the epilepsy group and two in the control group. One dog in the epilepsy group developed panting (i.e., very rapid and shallow breathing). Two dogs in the control group developed tachypnea (i.e., an increased respiratory rate). Panting and tachypnea were identified based on the dogs’ appearances and capnogram waveforms on the monitor. These changes appeared immediately when the total bemegride dose exceeded 8 mg/kg. Panting and tachypnea disappeared within a few minutes without any intervention. However, once these respiratory changes occurred, they recurred after every additional dose of bemegride. EtCO_2_ decreased and the respiratory rate increased in these three dogs while respiratory changes persisted. However, SpO_2_ and the heart rate were not affected. These changes were not observed in the remaining five dogs (3/4 dogs in the epilepsy group and 2/4 dogs in the control group). The EtCO_2_, SpO_2_, heart rate, and respiratory rate of the remaining dogs were unchanged and there were no arrhythmias during EEG recordings.

Phenobarbital and diazepam were administered to all eight dogs in accordance with the predetermined method. EDs disappeared immediately in all dogs with EDs after the injection of 3 mg/kg phenobarbital and 0.5 mg/kg diazepam.

A few minutes after awakening, one dog in each group vomited a small amount of gastric juice once.

The median total EEG recording times in the epilepsy and control groups were 45.5 (range: 32–55, IQR: 34.5–53.5) min and 67 (range: 65–70, IQR: 65.75–68.5) min, respectively. The concentration of sevoflurane in each dog during the baseline and activation periods is shown in Table 6.

All dogs were kept under observation for 24 h from the end of EEG recordings and there were no seizures. There were no abnormal findings in physical and neurological examinations or hematology and serum biochemistry tests the day after EEG recordings.

## 4. Discussion

The present study investigated the potential of bemegride as a pharmacological activation agent of EDs in dogs. Activation methods are techniques that enhance or provoke preexisting EEG abnormalities in patients with epilepsy. Bemegride, a central nervous system (CNS) stimulant and antagonist of barbiturates, has been used as a respiratory stimulant to treat barbiturate overdose and as an inducer of EDs or clinical seizures in human medicine on rare occasions [18,19,20,21,27]. It was previously shown to activate EDs in 67.5–82.7% of patients with epilepsy [19,20,21]. In the present study, bemegride enhanced or provoked EDs during EEG recordings under sevoflurane anesthesia in all dogs with epilepsy. EDs were provoked in dogs that did not show EDs prior to the administration of bemegride. When EDs were already present, bemegride increased their frequency. Although EDs were induced after the administration of bemegride in 50% of dogs without epilepsy, the dose required to induce EDs was significantly higher and EDs were less frequent than in dogs with epilepsy.

A previous study reported that bemegride activated EDs at a spontaneous epileptic focus, currently known as the irritative zone or seizure-onset zone, in humans [31]. The seizure-onset zone is the cortex area initiating clinical seizures [29] (pp. 137–159). In the present study, the estimated regions of irritative zones before and after the administration of bemegride were consistent. In one dog with epilepsy, EDs were not detected before the administration of bemegride, but were successfully provoked by bemegride. The region of the irritative zone identified after the administration of bemegride and that found when EEG was subsequently performed on another day without bemegride were consistent. These results indicate that bemegride activated EDs at the spontaneous irritative zone in dogs, similar to humans.

In the present study, the frequency of EDs was slightly increased by the administration of bemegride in dogs with epilepsy. The difference may not have been significant due to the small sample size. Alternatively, a mild increase in the frequency of EDs may be a clinical advantage when considering the safety of bemegride.

Regarding the dose required for activation, EDs were elicited with a markedly lower dose of bemegride in dogs with epilepsy (median, 7.3 mg/kg) than in those without (median, 19.7 mg/kg). Furthermore, there was no overlap in the required doses. This result satisfied the criteria of activation methods, namely, that the threshold difference between epilepsy and other patients needs to be large [20]. In humans, ED activation requires 20–120 mg of bemegride in patients with epilepsy and 150–200 mg in those without [19], reflecting a large threshold difference. Many conditions are causing recurrent episodic events that can be misdiagnosed as epileptic seizures. For example, paroxysmal dyskinesia sometimes mimic focal epileptic seizures [32]. The importance of finding interictal EDs to distinguish between paroxysmal dyskinesia and focal epileptic seizures was recently reported in Pomeranians [33]. Bemegride may be used to help to make a diagnosis of epilepsy based on the dose required to activate EDs, especially in situations where a strict diagnosis is required. Although further study with a greater sample size to find the cut-off value of bemegride dosage as an activator in epileptic dogs is necessary, there seemed to be enough separation of dosing between dogs with epilepsy and those without that a test of up to 12 mg/kg would be sufficient to distinguish dogs with epilepsy and those without.

To the best of our knowledge, there has been no study on the appropriate dosing of bemegride as an activation agent in dogs. Although bemegride has been used to treat barbiturate intoxication in dogs [26] (p. 348), there have been no toxicity studies on bemegride in this species. Therefore, we used bemegride dosing according to the method of EEG activation in humans with epilepsy. The maximum dosage of bemegride was set at 20 mg/kg because the suggested dose of the drug for the treatment of barbiturate intoxication in cats and dogs is 15–20 mg/kg [26] (p. 348).

A change in the respiration pattern was observed in three out of the eight dogs examined in the present study. Two dogs developed tachypnea and the remaining dog developed panting when the total dose of bemegride exceeded 8 mg/kg. Bemegride stimulates the respiratory center [26] (p. 348), and this may have caused tachypnea. When the other dog developed panting, its EEG pattern changed to a low-amplitude fast activity rhythm, which is an EEG pattern of waking. Bemegride also exerts a stimulant effect on the CNS [26] (p. 348). Therefore, panting may simply have occurred because the depth of anesthesia was decreased by this effect. The present results suggest that changes in the respiratory pattern (i.e., tachypnea or panting) and a reduction in the depth of anesthesia occur with the administration of bemegride, particularly at a dose exceeding 8 mg/kg.

We performed EEG recordings under general anesthesia using sevoflurane, which does not suppress EDs, in contrast to many other inhaled anesthetics [34,35]. Therefore, sevoflurane is often used during epilepsy surgery in humans [34,35]. EDs have also been observed in dogs with IE under sevoflurane anesthesia [36]. We designed the present study in consideration of the use of bemegride for epilepsy surgery in dogs. Therefore, sevoflurane was used in this study.

BS is a type of periodic discharge of EEG in which periods of large amplitude and high-frequency activities (bursts) alternate with periods of almost no activity (suppression). It is observed in hypoxic encephalopathy and other brain dysfunctions or during deep anesthesia. A previous study showed that the burst activities of BS included spikes and sharp waves in dogs regardless of whether the dog had epilepsy, similar to humans [33]. To unify the depth of anesthesia, the concentration of sevoflurane was maintained at the point at which BS had just completely disappeared from the EEG trace in all dogs. Our facility has standardized this method for EEG recordings under sevoflurane anesthesia.

Shortly after awakening from anesthesia, 25% (two) of the dogs examined vomited once. The adverse effects of bemegride reported in humans include tachycardia, nausea, dizziness, and seizures [20]. To the best of our knowledge, the only reported adverse effect of bemegride in dogs is seizures [25], and there are no findings showing that vomiting is an adverse effect of bemegride. However, based on the present results, vomiting needs to be considered when bemegride is administered. A previous anecdotal study on dogs showed that a seizure occurred in one of the two awakened dogs after the administration of bemegride [25]. In this study, 12 mg/head bemegride was administered over 3 min, which was a markedly smaller dose than that used in the present study. No seizures occurred in the present study despite the higher dosage used. This may have been because we used bemegride under general anesthesia. However, seizure warnings always need to be heeded when using bemegride. Since vomiting was the only suspected adverse effect of bemegride in the present study, the results obtained indicate that serious adverse effects are less likely to occur when bemegride is used in accordance with our procedure.

Several criteria must be met before activation methods are employed to increase the diagnostic yield of EEG in patients with suspected epilepsy: the threshold difference between patients with epilepsy and other patients must be large; EDs must be activated in each patient’s spontaneous irritative zone; the procedure needs to be simple; and there cannot be any undesirable adverse effects [20]. Our procedures using bemegride are likely to fulfill these criteria.

Nevertheless, the present study has several limitations. The sample size was small and needs to be increased in order to accurately assess the utility of bemegride as a pharmacological ED activator in dogs for clinical use. Furthermore, limited information is currently available on the appropriate dosage of bemegride for dogs. Therefore, since different dosages of bemegride may influence the results obtained, examining the appropriate dosage is essential. For example, an attempt to investigate whether the same results may be obtained with a smaller amount of bemegride may be preferable. Moreover, the present study only used dogs with focal epileptic seizures evolving into generalized epileptic seizures and the effects of bemegride in dogs with other seizure types were not assessed. EDs activated by bemegride are known to originate from the cerebral cortex [23]. Therefore, it currently remains unclear whether bemegride exerts its effects on patients with an irritative zone under the cerebral cortex (e.g., the thalamus). The possible region of the irritative zone needs to be considered in each patient prior to the application of this procedure. Another limitation is that this study only evaluated interictal EEG. The irritative zone, which is the cortex area generating interictal EDs, and the seizure-onset zone, which is the cortex area initiating clinical seizures, may not always be in the same region. Ictal EEG obtained by continuous EEG recordings with video monitoring allows for the verification of whether the cortex area generating EDs activated by bemegride and the spontaneous seizure-onset zone are in the same location. In addition, there is still no standard protocol for EEG recordings in veterinary medicine. We used sevoflurane in the present study. The effects of and risks associated with the administration of bemegride in dogs that undergo EEG recordings with other protocols remain unknown.

## 5. Conclusions

In the present study, bemegride activated EDs in all dogs with epilepsy. These EDs predominantly occurred in each dog’s spontaneous irritative zone. The dosage of bemegride required for ED activation was significantly lower in dogs with epilepsy than in those without, suggesting that this test can be used to distinguish epilepsy from non-epilepsy. No serious adverse effects, such as seizures, were observed. We concluded that bemegride has potential as a safe and effective pharmacological activation agent of EDs in sevoflurane-anesthetized dogs with epilepsy. The results of this study may help in the diagnosis of epilepsy and provide more options for the therapeutic planning of epilepsy, including presurgical evaluations, in dogs in the future.

## Figures and Tables

**Figure 1 animals-12-03210-f001:**
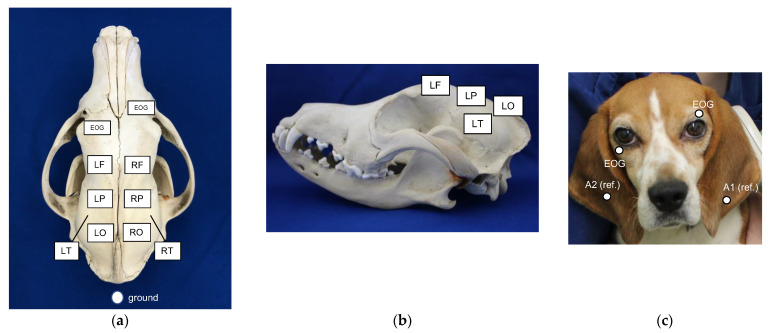
Electrode placement from dorsal (**a**) and lateral (**b**) views of the cranium and frontal view of a dog’s face (**c**). Recording electrodes were placed subcutaneously on both sides symmetrically over the frontal (LF, RF), parietal (LP, RP), occipital (LO, RO), and temporal (LT, RT) lobes. Reference electrodes were placed on the left and right ears (A1, A2). A ground electrode was placed subcutaneously on the dorsal of the neck above the atlas. Electrodes for recording eye movement were placed near the left and right eyelids (EOG).

**Figure 2 animals-12-03210-f002:**
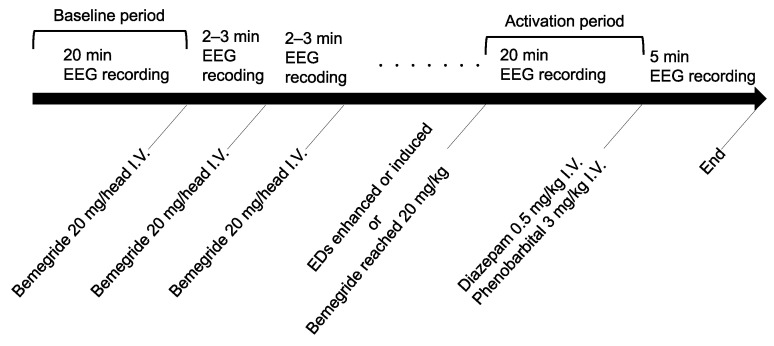
Timetable of bemegride administration and EEG recordings during the experiment. I. V.: intravenous injection.

**Figure 3 animals-12-03210-f003:**
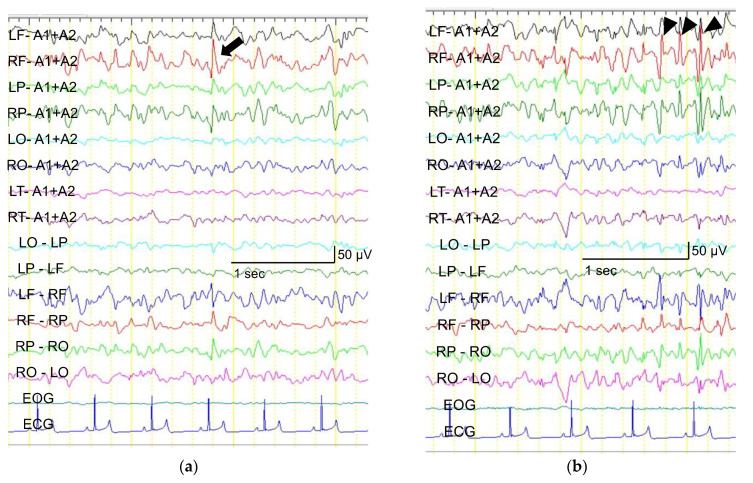
Screenshots of an electroencephalogram recording before (**a**) and after (**b**) the administration of bemegride in dog 1. The irritative zone was identified as the same region (i.e., right frontal) in the baseline and activation periods. An isolated sharp wave with a right frontal dominance (black arrow) was rarely observed (0.2/minute) before the administration of bemegride (**a**). A series of spikes (black arrowheads) newly appeared. Isolated spikes and a series of spikes with a right frontal dominance were more frequently noted (1.6/minute) after the administration of bemegride (**b**). Sensitivity: 10 µV, time constant: 0.3 s, high-frequency filter: 60 Hz, and notch filter: off.

**Figure 4 animals-12-03210-f004:**
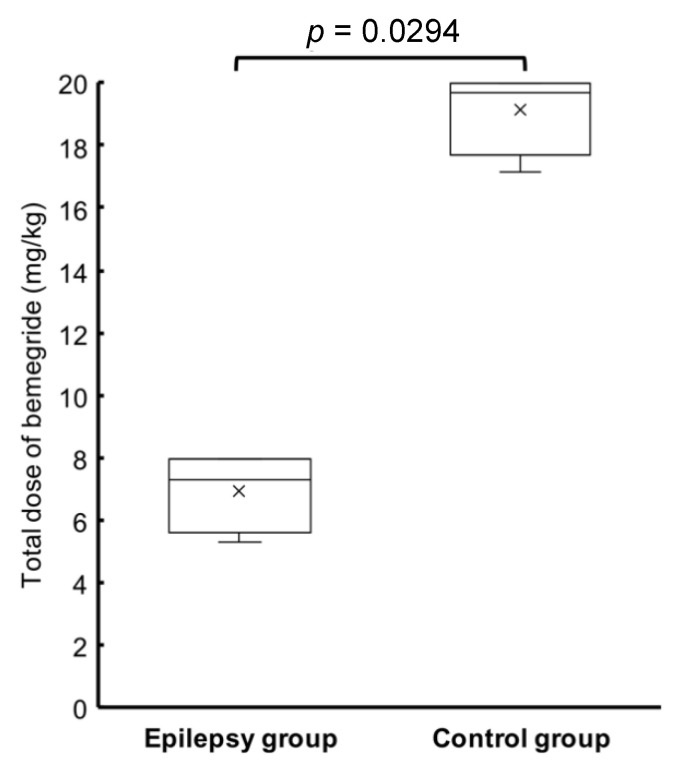
Comparison of the bemegride dose required to enhance or induce EDs between epilepsy and control groups. The vertical line shows the dose. The horizontal line in both boxes represents the median dose. The cross represents the mean dose. The vertical line below the box represents the first quartile of the data. The dose of bemegride was significantly lower in the epilepsy group than in the control group (*p* = 0.0294).

**Table 1 animals-12-03210-t001:** Information on dogs.

Dog	Age (Months)	Body Weight (kg)	Sex	Diagnosis	Seizure Frequency (per 3 Months)	Medication
Epilepsy group						
Pekinese (Dog 1)	25	3	Female	IE (Tier I)	2	KBr
Miniature Dachshund (Dog 2)	48	3.8	Female	IE (Tier III, necropsy)	4	PB, KBr
Beagle (Dog 3)	109	12	Female	IE (Tier I *, necropsy)	0.5	None
Beagle (Dog 4)	49	49	Male	IE (Tier I)	0.25	None
Control group						
Beagle (Dog 5)	60	10.45	Male			
Beagle (Dog 6)	25	9.3	Female			
Beagle (Dog 7)	26	10.3	Male			
Beagle (Dog 8)	60	10	Male			

Tier I *: Tier I criteria, except for urinalysis and serum NH_3_ levels. KBr: Potassium bromide. PB: Phenobarbital.

**Table 2 animals-12-03210-t002:** Seizures in each dog.

Dog	Seizure Type/Semiology	Laterality
Dog 1	Focal motor/facial twitching, then repeated jerking head movements, which rapidly progressed to generalized tonic clonic movements	No laterality
Dog 2	Focal motor/facial twitching that rapidly progressed to generalized tonic clonic movements	No laterality
Dog 3	Focal motor/head shaking, followed by clonic movements of the jaw, which rapidly progressed to generalized tonic movements	No laterality
Dog 4	Focal motor/facial twitching that rapidly progressed to generalized tonic clonic movements	No laterality

**Table 3 animals-12-03210-t003:** Comparison of the region of the irritative zone, at which EDs predominantly appeared, between baseline and activation periods in each dog.

Dog	Region of the Irritative Zone	
Epilepsy group	Baseline	Activation
Dog 1	Right frontal	Right frontal
Dog 2	Left parietal	Left parietal
Dog 3	None	Left parietal
Dog 4	Left parietal	Left parietal
Control group		
Dog 5	None	None
Dog 6	None	Left parietal
Dog 7	None	None
Dog 8	None	Left parietal

**Table 4 animals-12-03210-t004:** Comparison of the type of epileptiform discharges between baseline and activation periods in each dog.

Dog	Epileptiform Discharges	
Epilepsy group	Baseline	Activation
Dog 1	Sharp wave	Spike, sharp wave
Dog 2	Spike	Spike
Dog 3	None	Sharp wave
Dog 4	Sharp wave	Spike, rhythmic spikes
Control group		
Dog 5	None	None
Dog 6	None	Spike
Dog 7	None	None
Dog 8	None	Spike

**Table 5 animals-12-03210-t005:** Comparison of ED frequencies between baseline and activation periods in each dog.

Dog	ED Frequency (per Minute)		*p*-Value
Epilepsy group	Baseline	Activation	
Dog 1	0.2	1.6	
Dog 2	4.4	18.0	
Dog 3	0.0	1.0	
Dog 4	0.1	0.4	
			0.125
Control group			
Dog 5	0.0	0.0	
Dog 6	0.0	0.2	
Dog 7	0.0	0.0	
Dog 8	0.0	0.4	
			0.500

**Table 6 animals-12-03210-t006:** Sevoflurane concentrations during baseline and activation periods in each dog.

Dog	Sevoflurane Concentration (%)
Epilepsy group	
Dog 1	3.0
Dog 2	3.0
Dog 3	3.0
Dog 4	3.5
Control group	
Dog 5	3.5
Dog 6	3.0
Dog 7	3.5
Dog 8	3.0

## Data Availability

The data presented in this study are available in the article. Further data are available from the corresponding author on reasonable request.

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
