# Peer review of "The Potential of Bemegride as an Activation Agent in Electroencephalography in Dogs"

_animals, 2022, doi:10.3390/ani12223210_

Round 1
Reviewer 1 Report
Hirashima et al.
This is an interesting paper that showed the sensitivity of epileptic dogs to systemic bemegride compared to non-epileptic dogs. There were 4 dogs in each group (epileptic, non-epileptic) and the results with regard to the bemegride dose necessary to elicit spontaneous interictal discharges are clear.
Major:
My one concern is that there could have been a clearer proposal for how to use bemegride diagnostically in practice. The authors explore doses up to 20 mg/kg, but there seemed to be enough separation of dosing between the epileptic and non-epileptic animals that a test of up to 12 mg/kg would be sufficient to distinguish epileptic from non-epileptic dogs. Clearly, the goal is to aid in diagnosis of epilepsy, but it is not clear how one would approach a single dog and make a decision of epileptic or not.
Minor:
I am not clear what a 20 mg/head does means in terms of more commonplace language (e.g., mg/kg). If you mean 20 mg total at a time until a dose of 20 mg/kg (so a 5 kg dog could get 5 doses), then please explain this more clearly.
Reviewer 2 Report
The manuscript confirmed the potential of bemegride as an activation agent in electroencephalography in dogs. bemegride can directly excite the respiratory center and the vascular motor center, so that breathing increases, blood pressure rises slightly. The central stimulant effect is similar to pentathritine, which is antagonistic to barbiturates and other hypnotic drugs. The effect of the maintenance time is short, after intravenous injection can only maintain 10 ~ 20 minutes. The injection of bemegride is large and the speed is too fast, which can cause nausea and vomiting. Increased reflexes, muscle tremors and convulsions. Froscher, and Bulau (1984) reported that by bemegride could induce epileptic discharges by EEG in patients prone to epilepsy. The manuscript presents an additional function of bemegride for epileptic dogs under anesthesia. The electroencephalogram of epileptic dogs under anesthesia can be abnormally discharged and produce a typical electrocardiogram.The results of manuscript may help in the diagnosis of epilepsy and provide more options for the therapeutic lanning of epilepsy, including presurgical evaluations. The novelty of the manuscript is strong, and it is recommended to accept after minor revision for publication
1. Lack of Ethics committee approval and lot number.
2. The writing of the paper does not conform to the normal language expression, please revise it carefully, such as the last sentence of the paper:“....in dogs in the near future”
3. Added reference: W Fröscher, P Bülau EEG activation with bemegride in epilepsy diagnosis. 1: Literature review, EEG EMG Z Elektroenzephalogr Elektromyogr Verwandte Geb 1984 Jun;15(2):75-81.
